# Experimental Investigation of Suitable Cutting Conditions of Dry Drilling into High-Strength Structural Steel

**DOI:** 10.3390/ma14164381

**Published:** 2021-08-05

**Authors:** Lukáš Pelikán, Michal Slaný, Libor Beránek, Vladislav Andronov, Martin Nečas, Lenka Čepová

**Affiliations:** 1Department of Machining, Process Planning and Metrology, Faculty of Mechanical Engineering, Czech Technical University in Prague, 166 00 Prague, Czech Republic; Michal.Slany@fs.cvut.cz (M.S.); Libor.Beranek@fs.cvut.cz (L.B.); Vladyslav.Andronov@fs.cvut.cz (V.A.); M.Necas@fs.cvut.cz (M.N.); 2Department of Machining and Assembly, Faculty of Mechanical Engineering, VSB-Technical University of Ostrava, 708 33 Ostrava, Czech Republic; lenka.cepova@vsb.cz

**Keywords:** dry machining, dry drilling, wet drilling, S960QL machining

## Abstract

Dry machining is one of the main ways to reduce the environmental burden of the machining process and reduce the negative effect of the cutting fluid and aerosols on operators. In addition, dry machining can reduce overall machining costs and, in the case of large workpieces, reduce the extra work associated with removing residual cutting fluid from the workpiece and adjacent area. For high-strength structural steel products, it is typical to drill holes with larger diameters of around 20 mm. Therefore, this work is devoted to the investigation of the dry drilling process carried out on a workpiece made of S960QL steel with a helical drill with a diameter of 21 mm. The aim was to find suitable cutting conditions for dry drilling with regard to process stability and workpiece quality. An experiment performed with a coolant served as a comparison base. A dry drilling experiment was performed with cutting speeds from 30 to 70 m·min^−1^ and feeds from 0.1 to 0.3 mm·rev^−1^, and with the results of this experiment, the same experiment with flood cooling was performed. During the drilling process, spindle torque values were recorded using the indirect spindle current recording method. The macroscopic chip morphology was studied to understand the cutting process. The chip thickness ratio was measured, as well as the maximum diameter of spiral chips. On the final workpiece, the qualitative and dimensional parameters of the holes were evaluated, such as the diameter, cylindricity and surface roughness, depending on the change in the cutting conditions and cutting environment. Evaluation of the obtained data led to the following conclusions. When drilling the S960QL material, there is only a very small increase in the drilling torque during dry drilling compared to drilling with cutting fluid. The increase in friction demonstrated by the chip thickness coefficient is significant. The influence of the environment on the dimensional accuracy showed a tendency for a slight increase in the holes’ diameters during dry machining. In comparison, the cylindricity of the dry-drilled holes shows a lower deviation than the holes drilled with cutting fluid. The surface roughness of the holes after dry drilling is affected by the increased friction of the outgoing chips, despite the resulting parameters being very good due to the drilling technology standards. This work provides a comprehensive view of the dry drilling process under defined conditions, and the results represent suitable cutting conditions to achieve a stable cutting process and a suitable quality of drilled holes.

## 1. Introduction

Our goal should be to achieve sustainability in all areas of human activity. Sustainability in industry can be understood as an approach where development should not compromise the prospects of our future generations [1]. Sustainability, of course, deals not only with environmental and human benefits but also with economic benefits. This article, therefore, deals with the technology of dry drilling of high-strength structural steels. The application of dry machining technology generally raises a number of issues that need to be addressed. However, there are many strong arguments for approaching machining without cutting fluids. The environmental impact of using cutting fluids is a major reason for the application of dry machining. The problem is the recycling of these fluids and their return to the ecosystem. Water consumption is also becoming increasingly important as a basic raw material for cutting fluids. Water availability is worsening in many places on the planet, while the wheels of industry are spinning faster and faster [2]. The use of cutting fluids in machining processes also has a negative effect on the health of operators, causes skin eczema and, in the form of aerosols, spreads outside the machine and endangers the respiratory tract of operators [1,3,4,5].

Another aspect for the deployment of dry machining is the cost associated with the acquisition, operation and recycling of cutting fluids. These aspects alone are estimated at hundreds of billions of US dollars in total global machining costs, considering only the world’s largest economies [6]. The cost of fluids can enter the calculation of machining costs with a share of about 10–30% [1,3]. In the case of large parts with demanding handling, the removal of process fluids and their degreasing also contribute significant costs. The total costs associated with the use of cutting fluids should therefore include the costs of degreasing, workpiece handling and the associated extension of the production time.

The motivation for solving dry drilling technology on high-strength structural steel workpieces is the products for which these materials are typically used. These are large welded structures and frames for the construction of special heavy-duty trucks, mobile cranes and agricultural machinery. These products tend to have a number of peripherals that are attached to the main frames by screw connections. For these joints, holes and threads must be machined in the main frame. We often encounter joints of size M20 and larger. When clamping the welded frames into the machine tool, the machined peripherals are at a height of several meters. This escalates the above-mentioned problems with cutting fluid because it does not return to the machine fluid circuit, increases its consumption and endangers the health of operators in a wide range of machines, and it is necessary to remove fluid from multi-ton products and from the area adjacent to the machine. Weldox 960 (S960QL) was chosen as a representative of high-strength structural steel, which is often used for the production of special vehicles with a requirement for high durability and low weight. It is a fine-grained martensite-bainitic steel, which also falls into the category of hard-to-machine materials even when machining with cutting fluid [7]. At normal temperatures, this material has a minimum yield strength of 960 MPa. However, the mechanical properties of S960QL decrease faster with increasing temperature than, for example, HSS or mild steel. Already at 100 °C, the tensile strength decreases by 5%; at 300 °C, it decreases by 10%; and at 500 °C, it decreases by 25% [8,9]. These properties can thus prove to be an advantage in dry machining.

### 1.1. Specifics of Dry Machining Technology

When removing cutting fluid from the machining process, we usually encounter problems such as temperature increase at the cutting point, excessive tool wear or sudden destruction of a tool, problematic chip formation and its departure from the cutting point [1,2,3,4,10]. Another problem may be the dimensional specifications of the product influenced by the thermal expansion of the workpiece and tool material [11,12,13,14]. The quality of the machined surface can deteriorate because the cutting fluid has a positive effect on the formed surface from several points of view. Without cutting fluid, there is an increase in cutting forces, an increase in the friction coefficient and an increase in the chip thickness ratio, and due to the higher temperature, the primary plastic deformation zone changes. The effect of cutting fluid is also a stabilizing effect, and without cutting fluid, there are often problems with excessive vibrations [15,16,17,18].

Drilling is a specific technology. For successful dry drilling, it is necessary to ensure the stable formation of suitable chips and the ratios between the cutting forces in order to preserve the stability of the cutting process and the smooth departure of chips from the cutting point. The technology depends on a number of parameters such as the diameter, length, geometry and coating of the tool, drilling depth, behavior of the machined material under different cutting conditions, the environment or the rigidity of the machine–tool–workpiece system [11,14,19]. Due to the increased friction coefficient, the thrust force increases by 10–30% and the drilling torque by 20–80%, depending on the workpiece material and cutting conditions [20,21]. Due to the dependence of the stability and accuracy of drilling on the balanced ratio of cutting forces on individual cutting edges, geometric errors of the hole, such as cylindricity deviations or perpendicularity of the hole, can occur. Furthermore, the chip thickness ratio increases, which results in tensile stress changes in the chip and the initiation speed of its departure [22,23]. The surface roughness after drilling is primarily dependent on the cutting conditions, but it is also significantly affected by the cutting environment. The achieved results vary considerably in this field. Some sources report deterioration of surface roughness in the order of 10%; other sources report deterioration of up to 100% depending on the workpiece material and cutting conditions [10,21].

### 1.2. Determining the Direction of Research

Most experimental work in the field of dry drilling is focused on drilling holes of smaller diameters. Requirements for typical applications of the given category of material, high-strength structural steels, are for holes of larger diameters around 20 mm and larger. This brings significantly higher cutting forces, and the range of tools and their concepts also differ significantly. For diameters around 3–6 mm, on which most work focuses, the most common conception is a monolithic twist drill. In contrast to diameters of 20 mm and larger, the monolithic concept appears only rarely, and indexable head drills or exchangeable insert drills predominate. The difference is also in the geometry of the tools. The chisel edge has a significant effect on the thrust force, generating more than 50% of the total thrust force [22]. For larger diameter tools, a wide range of chisel edge concepts with very complex geometries can be found.

Dry machining of high-strength structural steel has only rarely been solved. For a successful application of dry drilling, it is important to know the force parameters of the cutting process, to understand the formation and morphology of chips and to know what qualitative parameters of the hole can be achieved. To this goal, the following experiments were performed, which describe the behavior of the cutting process under different cutting conditions.

## 2. Materials and Methods

### 2.1. Materials

The experimental study was carried out on general structural steel Weldox 960. Weldox 960 meets the requirements of the corresponding steel grade S960QL according to EN 10 025-2. The chemical composition of Weldox 960 is mentioned in Table 1 and mechanical properties are mentioned in Table 2. A typical application for this steel is demanding load-bearing structures where the strength and good weldability are used as advantages [24].

As a workpiece, various plasma-cut Weldox 960 sheets (SAAB AB, Stockholm, Sweden) were used. Since plasma cutting leaves a heat-affected zone of an area of about 0.5 to 1 mm from the cutting line, depending on the sheet thickness and cutting parameters [25], the drilled area was always shifted by at least 5 mm from the edge of the workpiece.

### 2.2. Equipment

As a machine tool for this experiment, Okuma MU—400V II (Figure 1) (Okuma, Nagoya, Japan) was used. It has a 5-axis milling center powered by a 22 KW spindle motor with maximum output revolutions of 15,000 rpm. This machine allows internal tool cooling with the HSK-A63 interface. As a cooling medium, cutting fluid or compressed air can be used. The machine was equipped with a control system, Okuma OSP-P300MA.

The Iscar Sumocahm DCN A—3D twist drill (Figure 2) (Iscar LTD, Migdal, Israel) with a diameter of 21 mm and an indexable head, ICP 210, was chosen as the testing tool. It is a drill with a 140° tool tip angle, 1% back taper and a helix angle of 25°. The tool has 2 channels for internal cooling. The indexable head is made of sintered carbide grade IC908, which is sintered carbide P20 combined with a TiAlN PVD coating.

This tool was selected based on previous tests [26,27], where its specific geometry proved to be very suitable for dry machining purposes.

### 2.3. Methods

The stability of the process and chip evacuation is important for understanding the process of drilling. We attempt to describe the process by parameters that can be quantified and that have a direct impact on the stability of the process and chip evacuation.

In this experiment, blind hole drilling tests with a diameter of 21 mm and a depth of 38 mm were performed. Drilling was perfomed on plasma-cut plates with a material thickness of 40 mm. The Okuma MU—400V II machine was used for this test. The cutting speed v_c_, feed rate f and cutting environment were chosen as variable parameters. The feed rate was chosen in the range f = 0.1 mm·rev^−1^, f = 0.15 mm·rev^−1^, f = 0.2 mm·rev^−1^, f = 0.25 mm·rev^−1^, f = 0.3 mm·rev^−1^. The cutting speed was chosen in the range v_c_ = 30 m·min^−1^, v_c_ = 40 m·min^−1^, v_c_ = 50 m·min^−1^, v_c_ = 60 m·min^−1^, v_c_ = 70 m·min^−1^. The range of cutting conditions was chosen based on previous tests [26,27] and the achieved cutting conditions in further research [14,28,29]. A full factorial experiment design was used, i.e., all combinations of feed rates and cutting speeds were tested in the experiment.

This test was performed in parallel with cutting fluid cooling and, without cutting fluid, only with the assistance of compressed air cooling through the center of the tool. The air pressure entering the machine was 6 bar. Due to losses and changes in the cross-section of the air line in the machine, the pressure at the spindle outlet was 4.5 bar. As a cutting fluid, 5% solution of Blaser Synergy 735 was used, which is a water-miscible, oil-free, full-synthetic metalworking fluid with neutral pH. The density of this fluid was 1.06 g/cm^2^, and its viscosity was 53 mm^2^/s. Cutting fluid was also supplied through the center of the tool.

When drilling, it is possible to monitor the torque signal, the thrust force (on the Z axis) and the deflection forces (forces acting on the workpiece on the X and Y axes). The torque is one of the most important variables in the drilling process [19]. For the purposes of this research, the torque on the machine spindle was monitored by an indirect method. Measurement of the spindle motor current is widely used as an indirect method of torque measurement [14,30,31]. In this case, the motor current was recorded as P_mot_. The cutting power P_c_ was recalculated from the spindle power P_mot_ using the efficiency characteristic of the spindle motor. For calculation of the torque Mc, equation 1 was used.
M_c_ = (P_c_ × 30 × 10^3^)/(π × n)(1)

Each measurement was performed at least 3 times, and the resulting values were calculated as an average value. Extreme values were excluded by indicative exclusion.

The chip thickness ratio was determined based on the ratio betwen the chip thickness and the thickness of the cut layer with a single cutting edge. The thickness of the cut layer was determined as half of the feed rate per revolution.
K = t_chip_/t_cut_(2)

The chip thickness was measured at several points in the middle chip area (Figure 3), and the values were averaged. Measurement was performed by an external micrometer gauge with conical tips.

Chip size was measured as the maximum diameter of a spiral chip (Figure 4) [32]. This is important information for assessing the ability of the chip to leave the cutting point. Chip measurements were always performed on at least 10 pieces of chips from each drilled hole and always in the same position. Measurement was performed by a double-scale sliding gauge.

The dimensions of the drilled holes were measured with a Carl Zeiss Prismo coordinate measuring machine (Carl Zeiss AG, Oberkochen, Germany) using a ruby touch probe with a diameter of 4 mm. The machine was equipped with an active scanning system that allows constant contact of the probe with the workpiece, and it scanned 200 points per second. The scanned data were processed using Zeiss Calypso software (6.4). A Gaussian filter with a filtration setting of 1-50 UPR was used to evaluate the diameters.

The surface roughness of the drilled holes was measured using a MarSurf LD 120 machine from the Mahr company (Mahr GmbH, Göttingen, Germany). Measurement was performed using probe arm Mahr LD B 4-10-2 7144 at 5 different locations. The measured data were processed using MarWin software (4.0). To evaluate the surface roughness, a basic wavelength λ_c_ of 0.8 mm was chosen, and thus the total measured length L_n_ was 4 mm. The surface scanning speed was v_t_ 0.5 mm/s.

## 3. Results and Discussion

### 3.1. Drilling Torque

The drilling torque during dry drilling into common mild steel increases by an average of 20–40% [2], but with this material, the increase in torque is significantly lower. The torque was recorded by the indirect method from the spindle current for the entire tested range of cutting conditions. The test was performed with compressed air cooling (Figure 5) and subsequently with cutting fluid (Figure 6).

The measured data show that when machining without cutting fluid, the torque increased by 4–7%, depending on the specific cutting conditions. Due to the absence of a lubricating fluid, significantly higher frictional forces are generated here [2,14]. However, due to the higher temperature at the cutting point, the cutting resistance of the machined material decreases, and thus the cutting forces decrease [1,3]. These two effects then negate each other, and, as a result, the resulting torque when machining without cutting fluid is only slightly higher.

In all cases, it is evident that the torque decreases with increasing cutting speed to the limit of 50 m·min^−1^. From this cutting speed, the torque value is essentially constant.

### 3.2. Chip Formation

The effect of the cutting speed on the chip thickness ratio has not been proven. Chip thicknesses at different cutting speeds, but at the same feed rate, were roughly 0.02 mm. The chip thickness ratio was evaluated only with respect to the feed rate.

The obtained values for the chip thickness ratio are within the common range for drilling technology, while the usual values are in the range 1.2–4 [22]. There is a significant difference in the values of the chip thickness ratio when machining under wet and dry conditions. The main reason is the significantly higher friction coefficient [2,16].

The chip thickness ratio not only provides us with a vision of the level of friction between the chip and the tool but it also allows us to determine the upper limit of the initial chip movement speed v_c.f_lim_.
v_c.f_lim_ = (ω × r_drill_)/K(3)
where *ω* is the drill rotation speed, r_drill_ is the drill radius, and K is the chip thickenss ratio.

For a cutting speed of 50 m·min^−1^ and a feed rate of 0.15 mm·rev^−1^, we then obtain the upper limit of the initial chip movement speed of 0.46 m·s^−1^ for wet machining and 0.34 m·s^−1^ for dry machining. The real initial chip movement speed is further dependent on the friction coefficient between the chip and the wall of the hole and drill flute [23]. This will increase the speed difference even more. Since chip evacuation during drilling depends on its initial movement speed, friction, tool geometry and depth of the cut, all these values, together, explain the poorer chip evacuation during dry drilling. Due to the trend of the graph in Figure 7, it follows that to reduce the compaction of chips in the drill flute, it is optimal to choose a higher feed rate, which allows a higher initial chip movement speed due to the lower chip thickness ratio.

The shape and size of the chip are important when it leaves the cutting point. When drilling with a twist drill, chips are formed in the typical spiral cone shape [22]. The outgoing chip must pass through a space defined by the flute of the drill and the wall of the hole. The deeper the drilling, the longer the path the chips have to travel. At the beginning, they have an initial velocity (see Equation (3)), and their velocity gradually decreases with the height of the elevation from the cut. The chips are decelerated mainly by friction. This friction takes place between the chip and the wall of the hole, and between the chip and the flute of the drill. The friction increases when larger chips are formed. The chip compresses like a spring in the flute, and the more it is compressed, the larger the friction. The resulting chip shape is also slightly formed during elevation [33]. Chip diameter dependence on cutting speed is shown in Figure 8 and Figure 9.

As a first result, we can say that in both cases, the chip diameter decreases with an increasing feed rate. The effect of the cutting speed is not conclusive. Based on the theory described above, a higher feed rate should aid in chip elevation because it aids in the formation of spiral chips with a smaller diameter. On the other hand, it should be mentioned that at higher feed rates, the chips are thicker and thus work as a stiffer spring when compressed in a drill flute, and even less compression can cause higher friction. As a second result, it can be said that dry machining brings, on average, a 0.5 mm larger chip diameter. This is mainly due to the lower friction coefficient during wet machining.

#### Chip Morphology

The results of individual measurements are presented by using graphs. Based on these data, conclusions on how cutting condition changes affect the cutting process can be drawn. The effect of these changes will be demonstrated on the macroscopic chip morphology for selected combinations of cutting conditions (Figure 10, Figure 11, Figure 12, Figure 13, Figure 14 and Figure 15). In drilling technology, it is also the shape of the chips that mainly affects their departure from the cutting point and the stability of the process [10].

Based on the trends of the chip thickness ratio and the chip diameter, it should be easier for the chips to leave the cutting point under a higher feed rate [34]. This conclusion was finally reflected during the tests, when it was really easier for the chip to leave the cutting point at higher feed rates. This parameter can be assessed primarily by observation and is therefore largely subjective. Although there are other ways to measure the level of chip departure from the cutting point [23], only the technique of qualified observational estimation was used in this experiment [10]. In Figure 14 and Figure 15, we can observe elementary chips with a less closed helix but, at the same time, a smaller diameter than in Figure 10.

Based on this observation and the macroscopic chip morphology, the conditions of poor chip separation and complicated departure from the cutting point were determined. These were generally all combinations of cutting conditions with the lowest feed rate of 0.1 mm·rev^−1^, but, partly, also with a feed rate of 0.15 mm·rev^−1^. At low feed rates, chip splitting does not occur because the thin chip is too malleable and there is insufficient tensile stress in the chip to cause it to split [22]. We can see snarled conical helical chips in Figure 11 and Figure 12.

Another problem area is the higher cutting speed in combination with the lower feed rate. From the point of view of chip separation, there is still the same problem with insufficient tensile stress. A higher cutting speed increases the temperature at the cutting point and thus causes a higher ductility of the chip, which requires higher tension for its splitting.

To solve the problem of chip splitting, it is advisable to move, with the cutting conditions, to the area where the chip naturally splits. If the choice of such cutting conditions is not possible for any reason, the solution may be to use a CNC drilling cycle with forced chip splitting. The principle is that after drilling a small depth increment (1.5 mm in this case), the drill retracts for a short moment by a small reverse step (0.3 mm). This will break the continuous chip. Then, drilling continues, and the whole process is constantly repeated. Every material and every type of tool needs to tune its own parameters of the drilling cycle with forced chip splitting. The graph in Figure 16 shows the trends of the drilling cycle that was applied in this test and successfully helped to split and remove the chips.

### 3.3. Dimensions

The measurement of drilled holes on a coordinate measuring machine took place at a depth of 10 mm, 19 mm and 28 mm. The measured data were processed using a Gaussian filter and averaged for the purpose of evaluating the hole diameter (Figure 17 and Figure 18).

From the results above, it can be seen that when drilling without cutting fluid, the average hole diameter is 0.03 mm larger than the diameter of the hole drilled with the cutting fluid. The effect of the feed rate on the size of the hole is also slight. The hole diameter increases slightly with an increasing feed rate: within the tested conditions, by 0.01 mm and less. Larger deviations can be observed during dry drilling. However, this may be due to the rougher surface of the dry-drilled holes, which affects the accuracy of the resulting measurement.

The quality of the drilled holes is mainly determined by the sum of the errors that occur during the drilling process such as tool tolerance and coaxiality, the imbalance of cutting forces, deviation of the hole perpendicularity, vibration, spindle clearance, thermal expansion of the tool and workpiece or the effect of outgoing chips on the hole wall. [35]. Another very important factor is the thermal expansion of the tool and workpiece material [12,35]. The most common cause of a hole shape error is dynamic deflections of the drill due to unbalanced forces [11]. Due to the very small increase in torque during dry and wet drilling, it can be assumed that the difference in dimensions, in this case, is mainly due to the thermal expansion of the tool substrate [15].

Focusing on the form errors of the drilled holes, the cylindricity of the holes was evaluated for this purpose (Figure 19 and Figure 20). The evaluation was performed using the three-section method.

From the dependence of the cylindricity of the holes on the cutting conditions, it is evident that the most regular hole is achieved with a higher cutting speed and a lower feed rate. Then, it is possible to reach a hole with a cylindricity of only 0.02 mm. On the contrary, the worst cylindricity is achieved with a low cutting speed and a higher feed rate. When machining with cutting fluid under a cutting speed of 30 m·min^−1^ and a feed rate of 0.25 mm·rev^−1^, the cylindricity of the hole is up to 0.09 mm. This trend can be explained by the effect of the torque. At the lowest cutting speeds and highest feed rates, the torque was the highest. The imbalanced forces on the individual cutting edges cause whirling. The imbalanced forces and their effect were more evident with higher absolute forces [11,16].

Furthermore, we can observe that the holes drilled with the cutting fluid reach a larger cylindricity error. In general, the cutting fluid should have a calming effect on the cutting process, and the achieved cutting forces should be lower [1,3,15]. Again, the obtained results confirm the theory that due to lower temperatures at the cutting point, the machined material shows a higher cutting resistance, and thus higher radial forces are achieved. These radial forces result in a lower accuracy of the drilled holes [1,13,15,16].

### 3.4. Surface Roughness

The roughness of the surface when drilling depends on a number of circumstances. The resulting surface roughness is influenced by the mechanical properties of the machined material, the geometry of the tool and its wear, the magnitude of the cutting forces and the stability of the process, the cutting environment and, last but not least, the cutting conditions. In the case of drilling, the feed rate should have the highest effect, followed by the cutting speed [4,17,18,36]. Drilling is a relatively specific machining process, and the depth of drilling also plays a role in the quality of the surface. The deeper we drill, the higher the risk of chip compaction in the flute, and this has a direct effect on the drilled hole quality. The surface roughness of drilled holes is shown in Figure 21 and Figure 22.

According to the surface roughness of the holes machined without cutting fluid (Figure 21), it is evident that the resulting surface is not directly dependent on the cutting conditions.

If we look at the surface of the dry-drilled holes in Figure 23, we can observe a significantly scratched surface by the leaving chip. This effect prevails over the other processes which form the surface of the holes. However, the resulting surface, mostly measured below Ra 1 µm, is relatively good due to the drilling technology standards [10,17,18,37].

When drilling with cutting fluid, the surface of the hole is formed mainly by tool marks, and the total values of the surface roughness reach even better parameters. With lower feed rates, the surface roughness of the holes is around Ra 0.6 µm. At a feed rate of 0.3 mm·rev^−1^, the surface quality deteriorates, as expected. From a feed rate of 0.25 mm·rev^−1^ and below, the surface quality no longer changes. The surface of the hole is more affected by other factors than the feed rate.

## 4. Conclusions

This paper presents a study of dry drilling technology applied to a high-strength structural steel S960QL workpiece. The aim of this study was to describe the process parameters, and to define the suitable cutting conditions and the achieved qualitative parameters of the drilled holes. As a comparison, the same tests with cutting fluid were also performed. A coated carbide indexable head twist drill with a relatively large diameter of 21 mm was used for testing. According to the experimental results, we can reach the following conclusions:A significant decrease in the spindle torque occurs when the cutting speed is increased to 50 m·min^−1^. From this value, the torque is basically constant. When drilling without cutting fluid, the torque is only about 4–7% higher compared to wet machining.The chip thickness ratio decreases significantly with an increasing feed rate. Additionally, the maximum diameter of the formed chips decreases with an increasing feed rate. The effect of the cutting speed on these values was not proven in this experiment. When evaluating the achieved results and the macroscopic morphology of the chips, the suitable cutting conditions for the dry drilling of the Weldox 960 material with the tool of the mentioned concept were determined. The cutting speed of around 50 m·min^−1^ reaches a suitable compromise between cutting forces, elemental chip formation and process stability. The feed rate of about 0.25 mm·rev^−1^ then reaches a suitable compromise in terms of splitting and chip size, chip thickness ratio, cutting forces and the quality of the drilled holes.From the point of view of the dimensional accuracy, it must be taken into account that the drilled holes are approximately 0.03 mm larger during dry machining. An interesting result was achieved. The cylindricity of the wet machined holes is slightly larger compared to that achieved with the dry-machined holes. The cylindricity increases with an increasing feed rate and a decreasing cutting speed, which is related to the magnitude of the torque and cutting forces. However, the cylindricity increases more when machining with cutting fluid. The surface roughness depends on the cutting conditions when machining with cutting fluid, and the resulting surface roughness can be achieved repeatedly. In dry machining, the surface roughness is primarily formed by the outgoing chips, and the resulting surface roughness is usually in a certain range (Ra 0.5–1.2 µm), but it is not possible to achieve it repeatedly.

## Figures and Tables

**Figure 1 materials-14-04381-f001:**
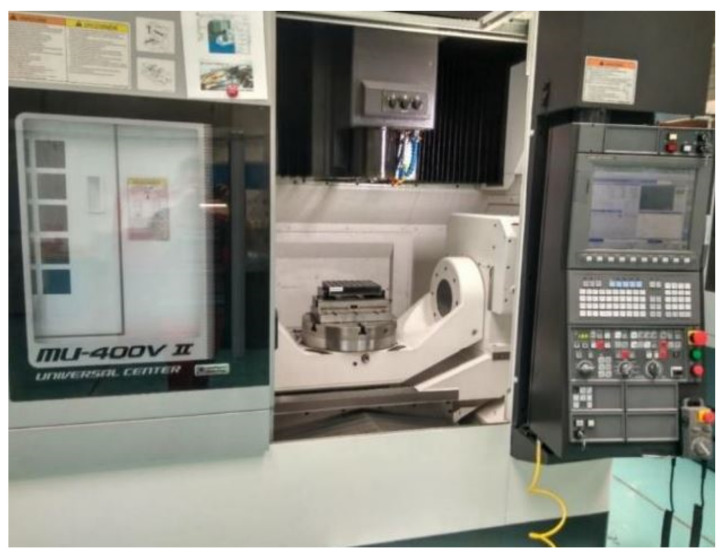
Okuma MU—400V II.

**Figure 2 materials-14-04381-f002:**
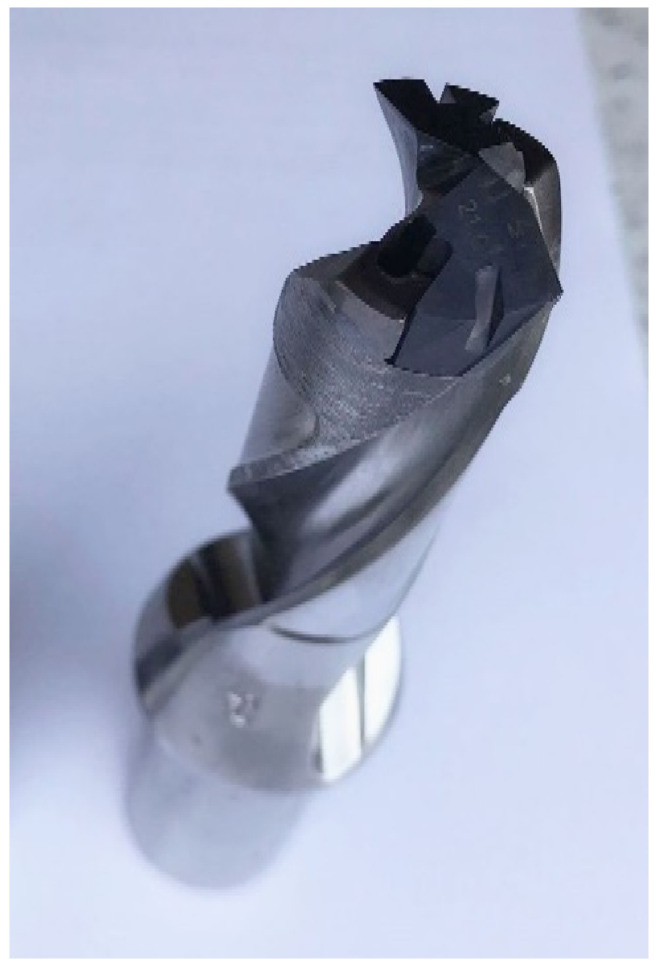
Drillbit Iscar Sumocham DCN-A.

**Figure 3 materials-14-04381-f003:**
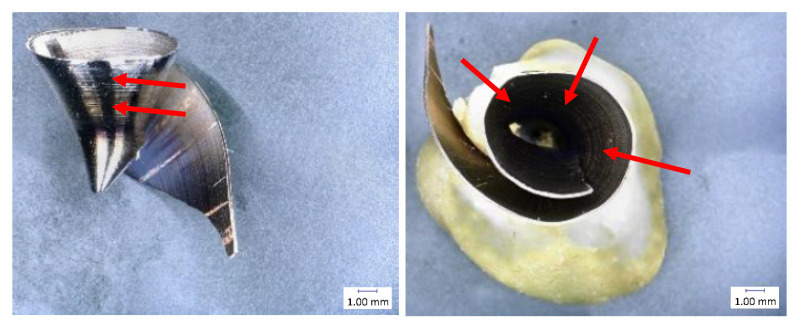
Measurement points of chip thickness.

**Figure 4 materials-14-04381-f004:**
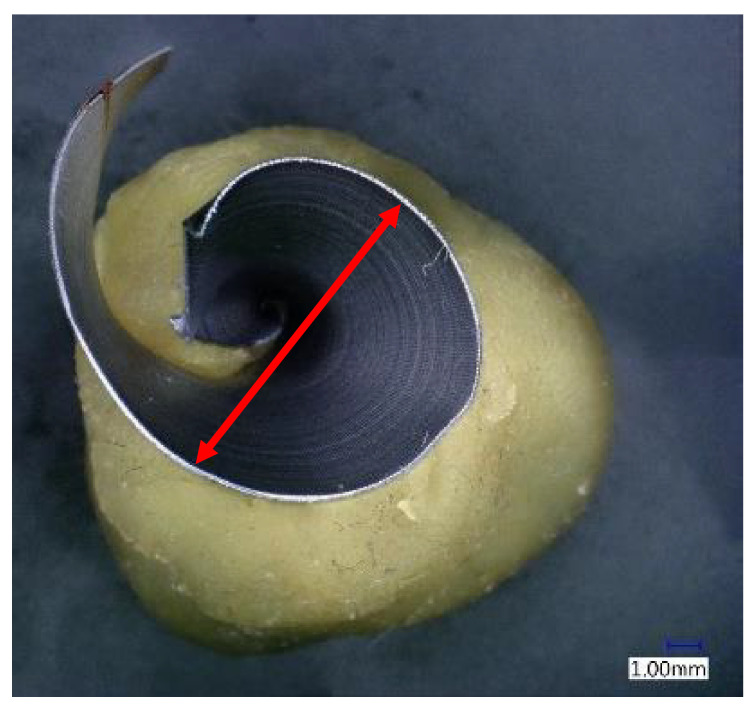
Measurement points of chip maximum diameter.

**Figure 5 materials-14-04381-f005:**
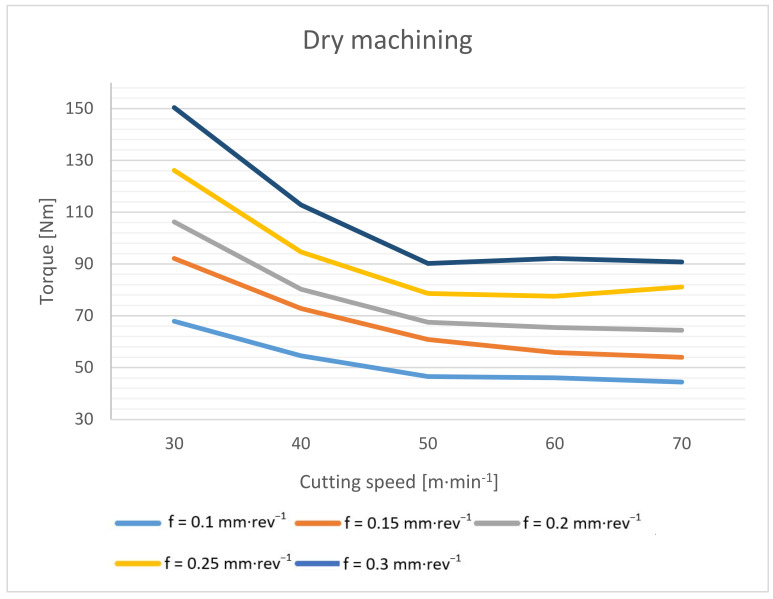
Drilling torque dependence on cutting speed and feed rate during dry machining.

**Figure 6 materials-14-04381-f006:**
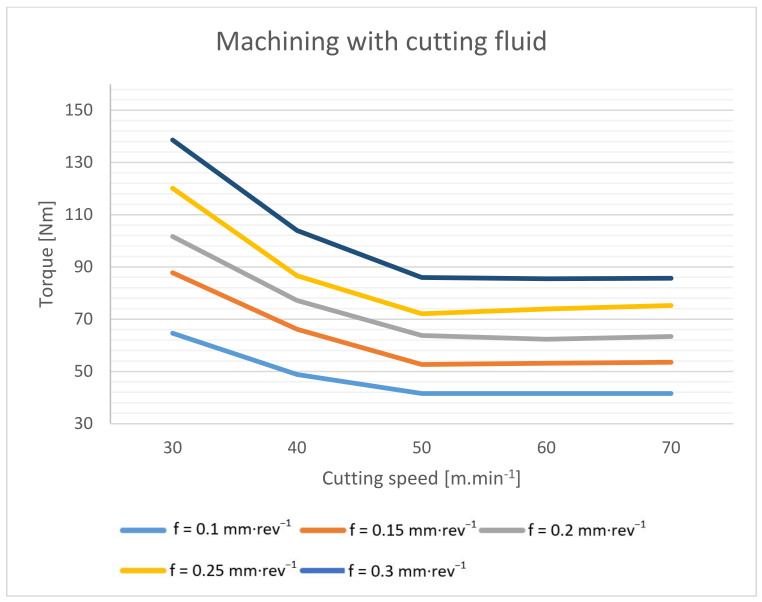
Drilling torque dependence on cutting speed and feed rate during machining with cutting fluid.

**Figure 7 materials-14-04381-f007:**
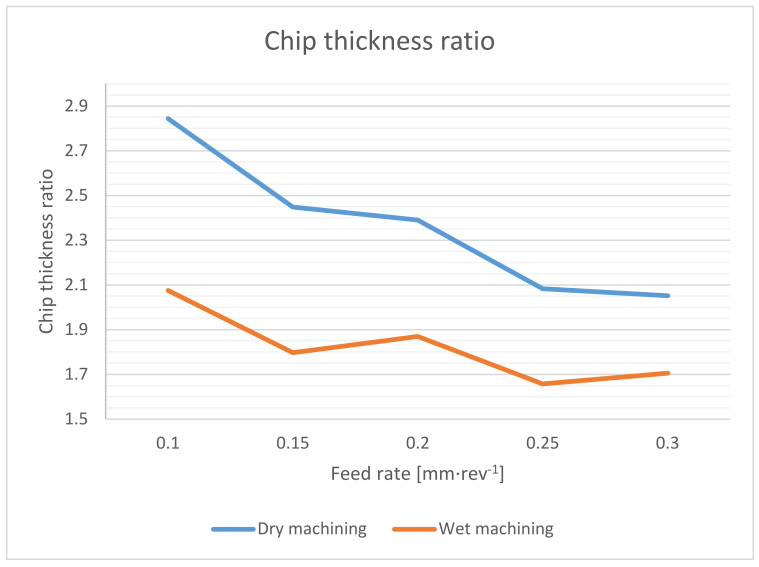
Chip thickness ratio dependence on feed rate during dry machining.

**Figure 8 materials-14-04381-f008:**
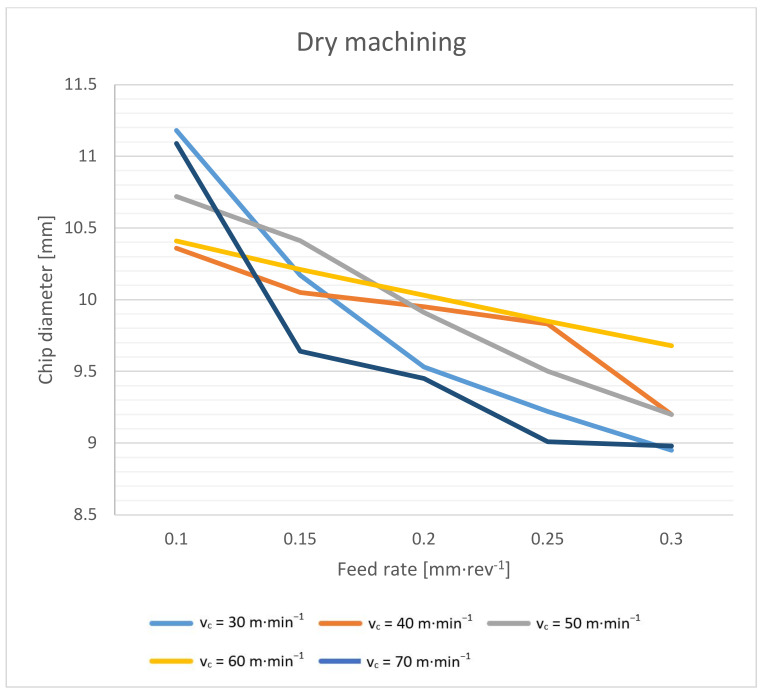
Chip diameter dependence on cutting speed and feed rate during dry machining.

**Figure 9 materials-14-04381-f009:**
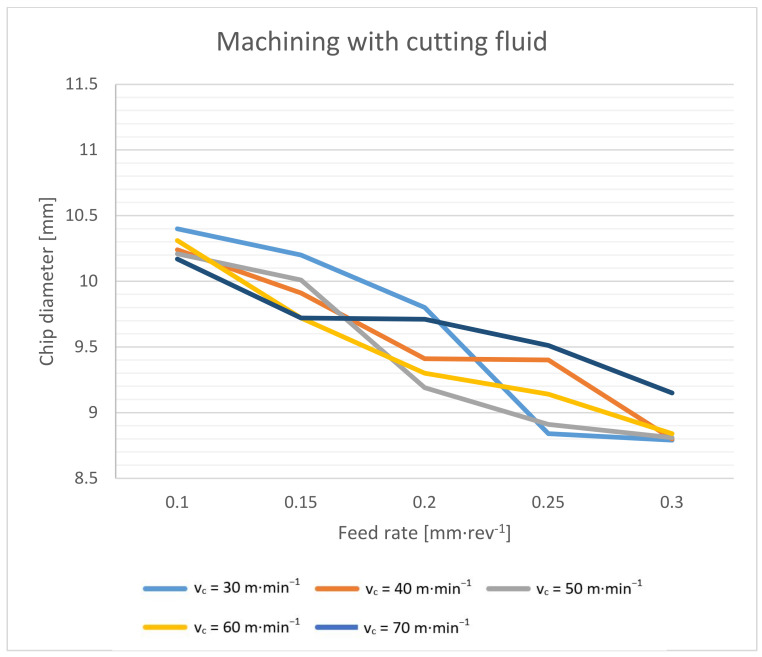
Chip diameter dependence on cutting speed and feed rate during machining with cutting fluid.

**Figure 10 materials-14-04381-f010:**
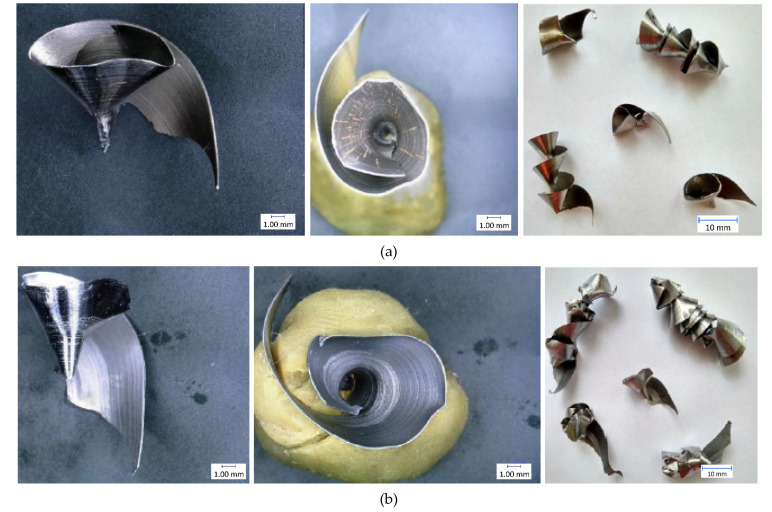
Chip morphology under conditions of v_c_ = 30 m·min^−1^; f = 0.1 mm·rev^−1^: (**a**) dry machining; (**b**) machining with cutting fluid.

**Figure 11 materials-14-04381-f011:**
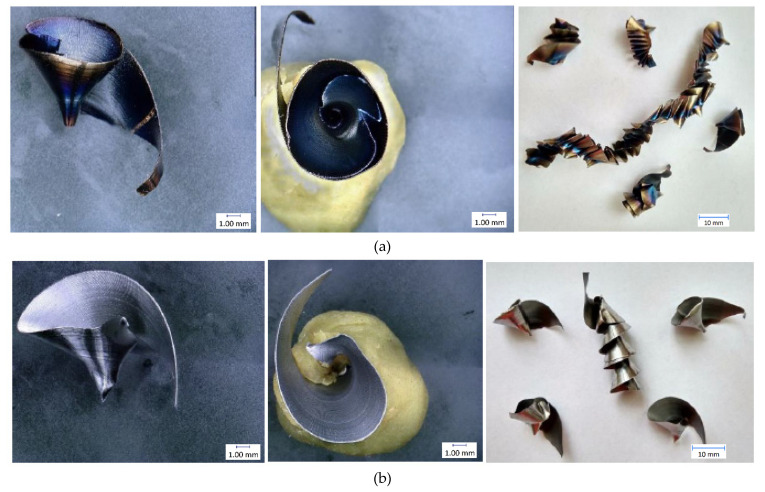
Chip morphology under conditions of v_c_ = 70 m·min^−1^; f = 0.1 mm·rev^−1^: (**a**) dry machining; (**b**) machining with cutting fluid.

**Figure 12 materials-14-04381-f012:**
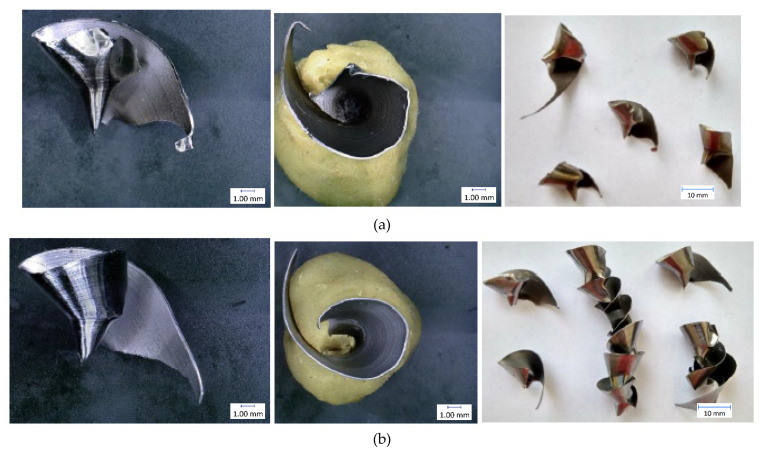
Chip morphology under conditions of v_c_ = 30 m·min^−1^; f = 0.2 mm·rev^−1^: (**a**) dry machining; (**b**) machining with cutting fluid.

**Figure 13 materials-14-04381-f013:**
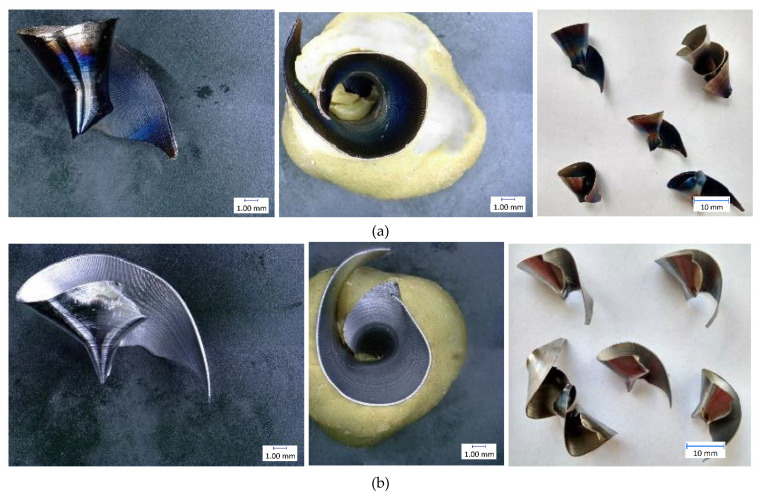
Chip morphology under conditions of v_c_ = 70 m·min^−1^; f = 0.2 mm·rev^−1^: (**a**) dry machining; (**b**) machining with cutting fluid.

**Figure 14 materials-14-04381-f014:**
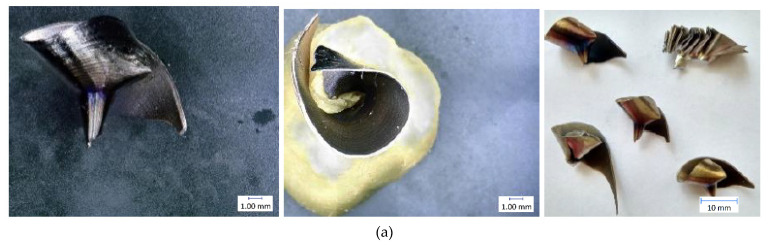
Chip morphology under conditions of v_c_ = 30 m·min^−1^; f = 0.3 mm·rev^−1^: (**a**) dry machining; (**b**) machining with cutting fluid.

**Figure 15 materials-14-04381-f015:**
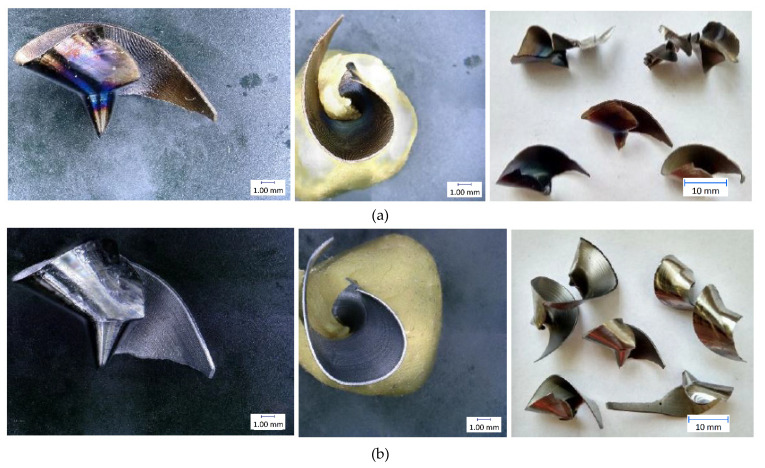
Chip morphology under conditions of v_c_ = 70 m·min^−1^; f = 0.3 mm·rev^−1^: (**a**) dry machining; (**b**) machining with cutting fluid.

**Figure 16 materials-14-04381-f016:**
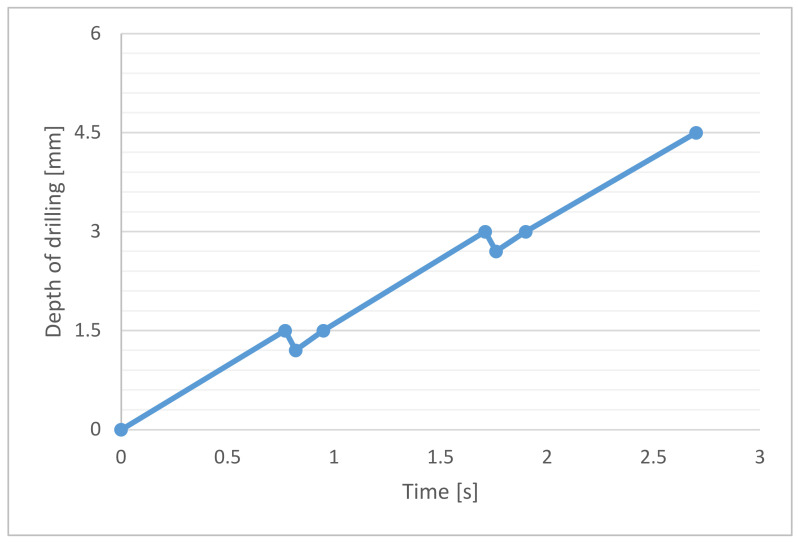
Timeline of forced chip breaking drilling cycle.

**Figure 17 materials-14-04381-f017:**
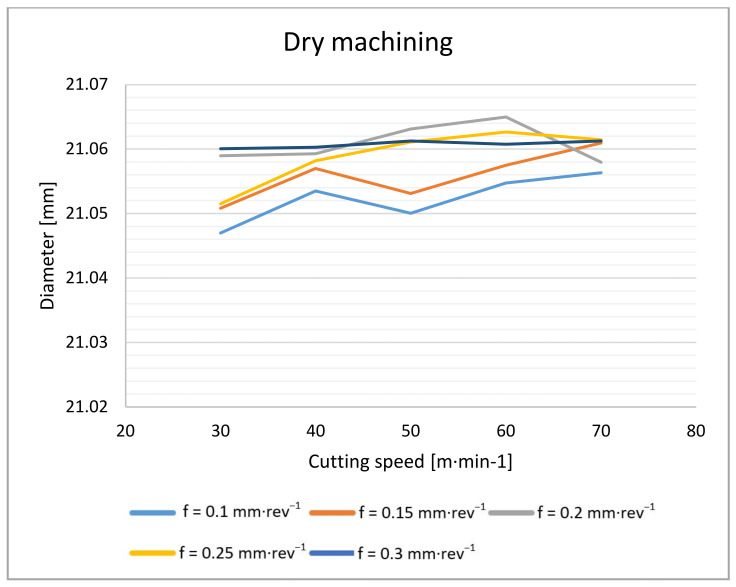
Hole diameter dependence on cutting speed and feed rate during dry machining.

**Figure 18 materials-14-04381-f018:**
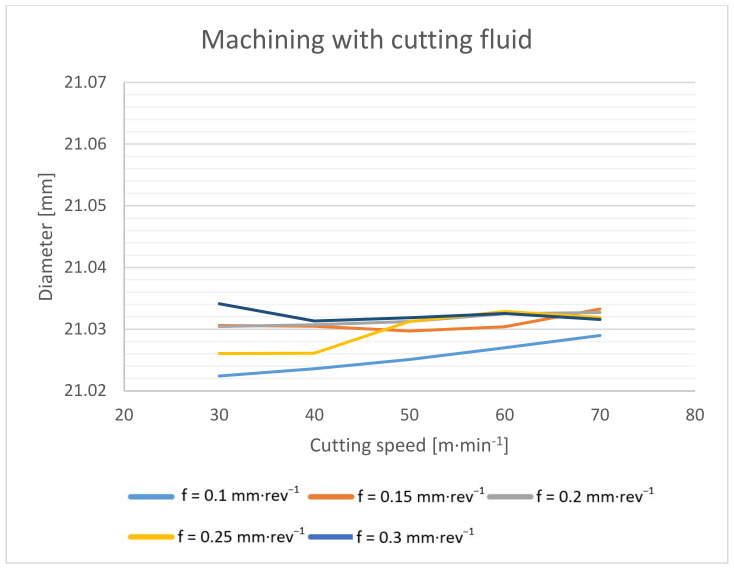
Hole diameter dependence on cutting speed and feed rate during machining with cutting fluid.

**Figure 19 materials-14-04381-f019:**
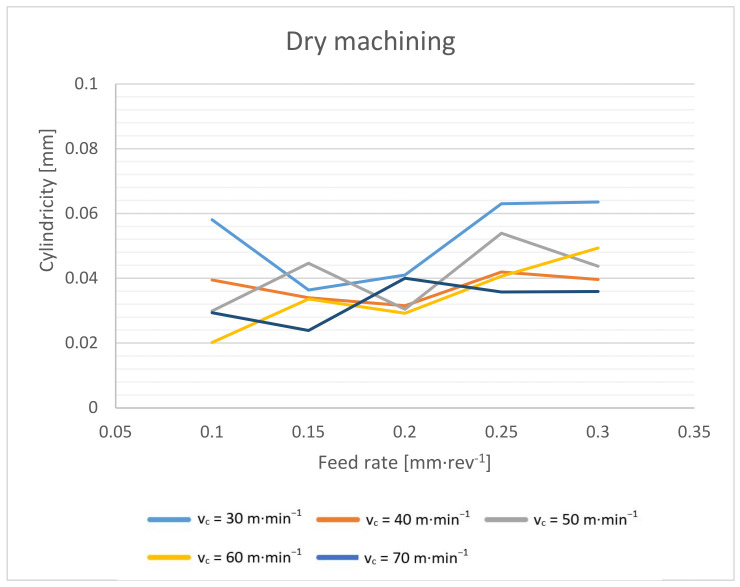
Cylindricity dependence on cutting speed and feed rate during dry machining.

**Figure 20 materials-14-04381-f020:**
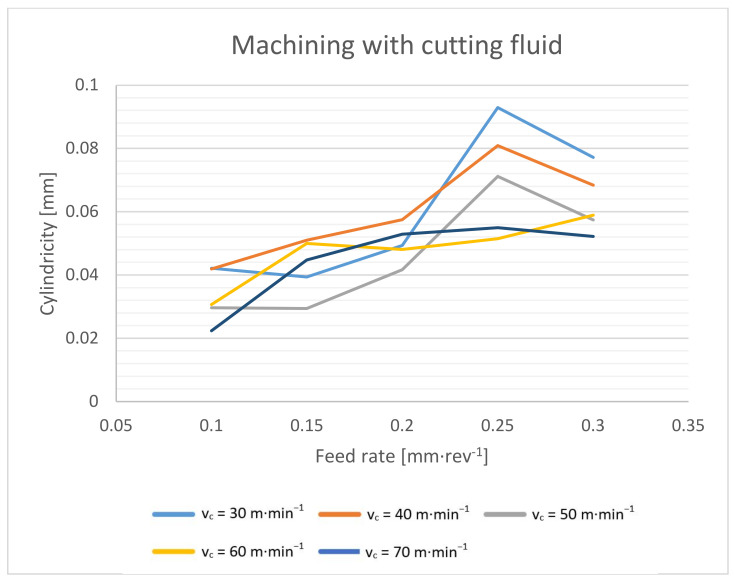
Cylindricity dependence on cutting speed and feed rate during machining with cutting fluid.

**Figure 21 materials-14-04381-f021:**
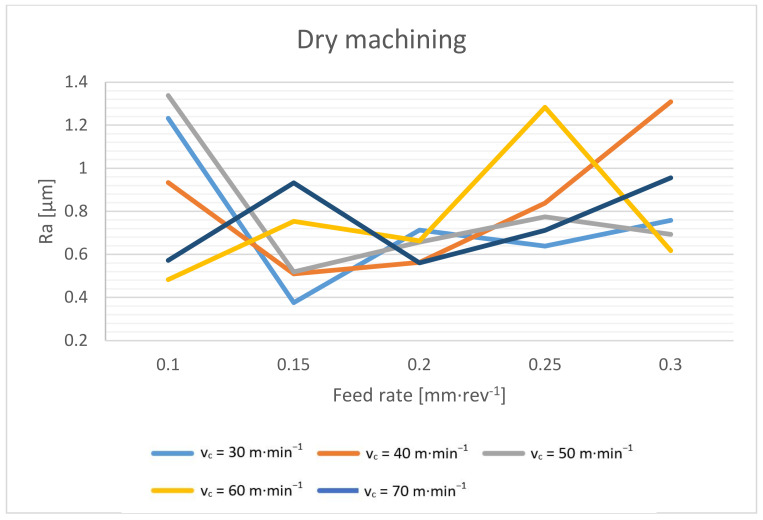
Surface roughness dependence on cutting speed and feed rate during dry machining.

**Figure 22 materials-14-04381-f022:**
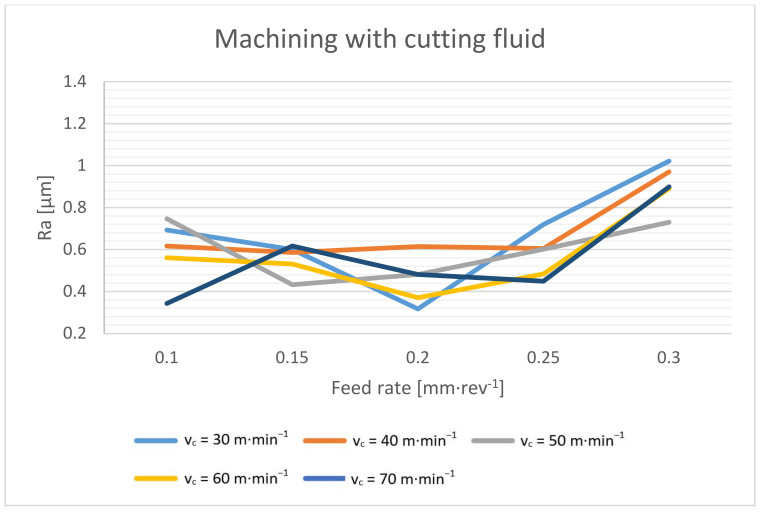
Surface roughness dependence on cutting speed and feed rate during machining with cutting fluid.

**Figure 23 materials-14-04381-f023:**
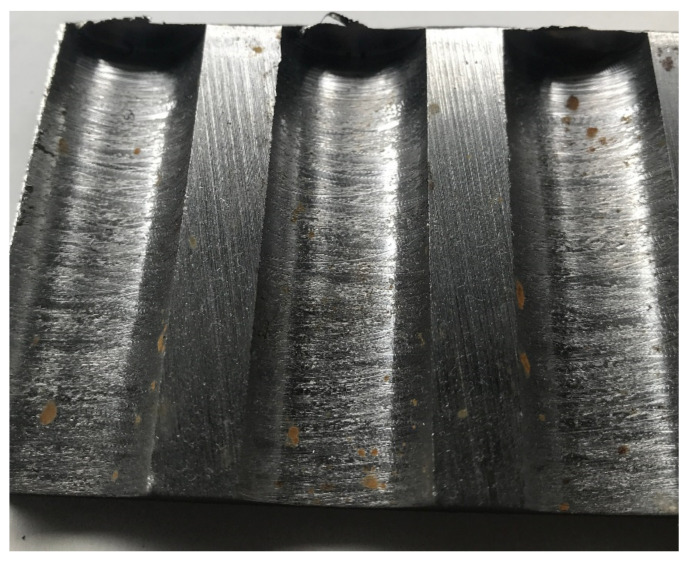
Dry machined hole surface in section.

**Table 1 materials-14-04381-t001:** Weldox 960 chemical composition.

C (max %)	Si (max %)	Mn (max %)	P (max %)	S (max %)	Cr (max %)	Cu (max %)	Ni (max %)	Mo (max %)
0.2	0.5	1.6	0.02	0.01	0.8	0.3	2.0	0.7

**Table 2 materials-14-04381-t002:** Weldox 960 mechanical properties.

Plate Thickness (mm)	Yield Strength R_p0.2_ (min MPa)	Tensile Strength R_m_ (MPa)	Elongation A_5_ (min %)
4–53	960	980–1150	12
53.1–100	850	900–1100	10

## Data Availability

Data are contained within the article.

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
