# Peer review of "Experimental Investigation of Suitable Cutting Conditions of Dry Drilling into High-Strength Structural Steel"

_materials, 2021, doi:10.3390/ma14164381_

Round 1

Reviewer 1 Report

the authors have defined their research more from the sustainability point of view and planned the experimental research on dry drilling. But they have actually done a very good job of examining the complete process of dry drilling and provided sufficient information that would be useful for other researchers who can take from there. As I said in my comments, the authors have not formulated the research problem completely as their interest seems to be only to provide dry drilling as a method to achieve sustainability. Though that is a good approach, there are better approaches that the authors could have adopted during the problem formulation stage. Unfortunately they could not do that and hence the paper resulted with that particular limitation. However the overall quality of the paper is excellent. The structural steel material that the authors have used is not regularly covered in the research and therefore would be a useful addition to the literature without any revision in my opinion.   Actually I did not suggest the revision of the paper in those lines, because that will be a major undertaking for the research to organize a Minimum Quantity Lubrication (MQL) using synthetic esters that are tailored for the work material to get a better performance compared to dry cutting. This could be a future project. I am giving a few examples here for such an idea: 

  1. Duc, T. M., & Chien, T. Q. (2019). Performance evaluation of MQL parameters using Al2O3 and MoS2 nanofluids in hard turning 90CrSi steel. Lubricants7(5), 40.
  2. Chinchanikar, S., & Choudhury, S. K. (2014). Hard turning using HiPIMS-coated carbide tools: Wear behavior under dry and minimum quantity lubrication (MQL). Measurement55, 536-548.
  3. Pal, A., Chatha, S. S., & Sidhu, H. S. (2021). Performance Evaluation of Various Vegetable Oils and Distilled Water as Base Fluids Using Eco-friendly MQL Technique in Drilling of AISI 321 Stainless Steel. International Journal of Precision Engineering and Manufacturing-Green Technology, 1-20.

Reviewer 2 Report

The article is very well presented. Both the material and the study method used are correctly explained. This article is of great scientific interest, since structural steel is a widely used material both in construction and in industry. The graphs obtained for dry cutting and cutting with refrigeration are very interesting. The photographs of the chips are of great quality. The low increase in cutting torque when using dry cutting, compared to cool cutting, is striking. As for the dimensions of the drilled holes, if you see more deviation. The great friction in dry cutting, if the roughness of the drilled hole increases. Its industrial application is very possible, seeing the results obtained. It is of sufficient quality to be published. I accept this article as it has been sent, it does not need any revision.

Reviewer 3 Report

Abstract: Please arrange the abstract according to scheme: introductory sentence, aim, the list of research, research results and highlight the novelty

Keyword: "dry and wet machining comparison" for two reasons is not appropriate:
1. Dry and wet machining comparison is the aim of the work?: If yes, why comparison is mentioned only in the abstract and conclusion.
2. Using five words as a keyword is not common.

Line 172: "All combinations of feed rates and cutting speeds were tested in the experiment" - please shortly describe the design of experiment performed (what kind of experiment was used, what was the goal of the experiment ... ?).

Line 326: coordinate measuring machine (please add the type of machine and producer).

Line 338: The quality of the drilled holes is mainly determined by the sum of the errors... (a citation is required) and better explained.

References should be renumbered. In the text, start with the number [1] and not [8].

Reviewer 4 Report

In this submission experimental investigation of dry drilling into high strength structural steels is described. This process is actual, however, some remarks should be made:

  1. It was said in Title that structural steels will be described; however only one steel have been studied; one steel using does not allow to evaluate the overall relationships of dry drilling,
  2. The standard EN 10 025 should be mentioned fully,
  3. Using spindle power data to determine cutting torque causes significant errors; it should be used a drill dinamometer for this purpose,
  4. The study does not say a word about the statistical treatment of the measurement results,
  5. The Fig. 3 (right) is not informative,
  6. What cutting fluid was used? And why only dry cutting is mentioned in Title?
  7. The relationships should be calculated and the corresponding graphs shown in Figs 5–6,
  8. Figs 10-15 should be shortened and described according ISO 3685:1993 standard,
  9. How was the data for fig. 16 calculated?
  10. The tolerances at the different hole locations should be specified (it can be seen from Figs 17 – 20 that feeds and speeds have the little effect),
  11. Roughness tests only confirmed that drilling is a rough machining process; no significant difference in roughness is seen under different drilling conditions.

Reviewer 5 Report

General Evaluation (Scope and Value)

The paper presents a challenging experimental comparison of drilling process, between dry and wet cutting conditions of a high strength structural steel. The study focuses on torque, chip dimension and roughness measurements, for several sets of cutting speed and feed rate combinations. The study presents major interest regarding the scale of machining process performed and especially the large size of drilled holes in combination with the difficult-to-cut material, such as a high strength structural steel workpiece. However, the elaborated investigation could be further improved by taking into consideration the following topics:

  • The study is a partial comparison between dry and lubricated cutting conditions; the effects on workpiece material integrity and cutting tool endurance were not investigated (the title of the paper has to be adapted).
  • The cutting fluid properties are not commented. This is important information since the study was based on the specific cutting fluid use as a lubricant.
  • The mechanisms of cutting were not studied in detail, concerning, for instance, chip microstructure, fracture mechanisms, plasticity and shear band formation, etc.
  • The heat extraction during cutting was not quantified.

Therefore, the paper seems that lacks of completeness in the frame material-process interaction, although it focused diligently on the macroscopic description of cutting operation and chip formation between dry and lubricated conditions. From the above general evaluation, the paper needs to be reconsidered in terms of completeness and scope fulfillment shedding more light in the field of materials and process relationships.

Specific Comments (Quality of Presentation)

  • The References have to be cited in the order of numbering.
  • The images used to show the chip formation and morphology have to be further improved in terms of quality and resolution. The pictures also are presented with extended background principally with blue color which makes a weak contrast with the main object.
  • Also appropriate scale markers are necessary even approximate for macroscopic images.
  • The captions are very lean; the use of indices (a), (b) in isolated pictures is suggested.
  • The chip thickness ratio (K) is not clearly defined; chip of cut layer has to be explicitly referred.
  • Measuring device for chip size has to be referred in 2.3 Section.
  • The manuscript is very narrative and complex at some sections (e.g. chip formation and chip morphology) and Figures 10-15 are not commented. Also other Figures, e.g. Figs. 8, 21 and 22 etc., seem that they were not either referred in the text.
  • The colors used in graphs have to be revised; especially the two different nuance blue colors cannot be easily distinguished.
  • There are several but minor language edits; please check the text carefully. For instance “cut” instead of “cuted” and “spindle” instead “spindel” (see Section 2.3), “common” instead of “common” (Section 3.2).

Round 2

Reviewer 4 Report

I agree with corrections made

Reviewer 5 Report

The manuscript has been revised and the majority of the review comments have been addressed. However, there are some additional questions, based on the authors' responses as follows:

  1. Basic lubricant properties were not referred although the trade name was included in the manuscript.
  2. Diagrams can be further improved, e.g. by better defining the Y-axis range, showing data-points and increasing significantly their readability. 
  3. Correction of minor spelling errors; see for instance L.194 ("...feed rate per 1 revolution" instead "... revoliton")
